# Thermoplastic Hybrid Composites with Wood Fibers: Bond Strength of Back-Injected Structures

**DOI:** 10.3390/ma15072473

**Published:** 2022-03-27

**Authors:** Frederik Obermeier, Peter Karlinger, Michael Schemme, Volker Altstädt

**Affiliations:** 1Department of Plastics Technology, Faculty of Engineering Sciences, Technical University of Applied Sciences Rosenheim, Hochschulstraße 1, 83024 Rosenheim, Germany; peter.karlinger@th-rosenheim.de (P.K.); michael.schemme@th-rosenheim.de (M.S.); 2Department of Polymer Engineering, University of Bayreuth, Universitätsstraße 30, 95447 Bayreuth, Germany; altstaedt@uni-bayreuth.de

**Keywords:** biocomposites, polypropylene, spruce, wood fibers, natural fibers, back-injection molding, bond strength

## Abstract

Due to their lightweight potential and good eco-balance, thermoplastic hybrid composites with natural fiber reinforcement have long been used in the automotive industry. A good alternative to natural fibers is wood fibers, which have similar properties but are also a single-material solution using domestic raw materials. However, there has been hardly any research into wood fibers in thermoplastic back-injected hybrid composites. This article compares the bond strength of an injection molded rib from polypropylene (PP) and wood fibers to different non-wovens. The non-wovens consisted of wood fibers (spruce) or alternatively natural fibers (kenaf, hemp), both with a polypropylene matrix. Pull-off and instrumented puncture impact tests show that, given similar parameters, the natural and wood-fiber-hybrid composites exhibit very similar trends in bond strength. Further tests using viscosity measurements, microscopy, and computed tomography confirm the results. Wood-fiber-reinforced thermoplastic hybrid composites can thus compete with the natural fiber composites in terms of their mechanical behavior and therefore present a good alternative in technical semi-structural applications.

## 1. Introduction

Biocomposites made of thermoplastic or thermoset matrices with wood (e.g., spruce) or other natural fibers (e.g., flax, hemp, kenaf) have long been applied in the automotive industry [1,2,3,4] because of their lightweight potential, shorter cycle times, and better eco-balance [5]. The main applications are in the automotive interior, e.g., door panels and instrument panels (see Figure 1a). Thermoplastic matrices are becoming more prevalent than the previously standard thermoset matrices because thermoplastic, unlike thermoset matrices, can be used in hybrid processes such as back-injection molding [5]. While natural fibers have long been used in thermoplastic matrices and hybrid processes, wood fibers have not yet been researched for hybrid processes such as back-injection molding. Recent review articles do not mention hybrid processing methods or back-injection of wood fiber composites [6,7,8,9,10,11].

An important property of hybrid composites is the bond strength (see Figure 1b) between flat semi-finished products (non-wovens) and back-injected structures (ribs, screw domes, edge areas, etc.). This article compares the bond strength of non-wovens with wood fibers (spruce) and alternatively non-wovens with natural fibers (kenaf, hemp) in thermoplastic hybrid composites. The comparison uses wood fiber non-wovens and natural fiber non-wovens with a polypropylene matrix. The non-wovens are back-injected with direct-compounded, wood-fiber-reinforced polypropylene using an injection-molding-compounder. The procedure of direct compounding of wood fibers for ribs or screw domes is relatively new [12,13,14,15]. The specimens made of the non-wovens and back-injected ribs are produced varying the following parameters: core temperature of the non-wovens, injection material, and rib geometry. Pull-off tests and instrumented puncture impact tests were used to determine the bond strength of the different specimens. The tests show that thermoplastic hybrid composites with wood fibers exhibit slightly lower bond strength compared to composites with natural fibers, but the trends in mechanical behavior of the two are very similar.

Various publications provide a good overview of previous applications of composites with wood or natural fibers. They primarily deal with applications in the automotive industry [17,18,19]. Compression molding has mainly used thermoset matrices and fiber materials such as wood or certain natural fibers (flax, hemp, kenaf, sisal, jute, and cotton). Wood fibers in the form of non-wovens and needle felts have been used in compression molding with both thermoset and thermoplastic matrices [20]. Fibrowood is a needled wood-fiber non-woven that uses acrylic resin and thermoplastic fibers as a matrix. The Wood-Stock process uses extruded polypropylene sheets containing wood flour and natural fibers [21]. Lignotock consists of wood fibers, acrylic resins, melamine resins, and thermoplastic fibers in various mixing ratios [22]. Englund et al. state that the use of wood fiber composites is widespread, but there is little scientific work on them. The literature predominantly deals with composites with wood chips and thermoset matrices [23]. Current research in semi-finished products aims to combine natural or regenerated cellulose fibers with glass, carbon, and basalt fibers to improve mechanical properties [24,25,26,27]. Another field of research is the use of biopolymers, biodegradable plastics [28,29,30,31,32,33,34], as well as recycled plastics [35,36,37,38]. Flame retardants for composites with wood fibers and polypropylene were investigated by Ayrilmis and his colleagues. They found that an optimum of physical and mechanical properties is achieved with 4 wt.-% coupling agent and 8 wt.-% flame retardant based on phosphate [39]. According to Renner et al., ammonium phosphate, graphite, metal hydroxide derivatives, and their combinations are very effective flame retardants for composites containing wood fibers [40]. Saba et al. review flame retardants for composites with kenaf fibers. In addition to Renner et al., they recommend montmorillonite, nanoclays, and nanotubes as flame retardants [41]. Hu et al. synthesized a new lignin-based flame retardant with phosphorus for biodegradable composites with wood powder and polylactic acid. In addition to improved flame retardancy, they were also able to demonstrate promising mechanical properties [42].

Hybrid processing methods, such as back-injection, are used for thermoplastic semi-finished products made from natural fibers (e.g., kenaf, hemp). In hybrid processes, the semi-finished product is first heated in an infrared oven or a hot press, then inserted into the injection mold, formed, and finally back-injected [43]. Hybrid processing methods profit from the synergy of the combination of both compression and injection molding. Back-injection molding combines the inherent stiffness of fiber-reinforced semi-finished products in compression molding with the design stiffness and high productivity of injection molding [44]. The development of hybrid processes for natural fiber-reinforced semi-finished products has been driven by suppliers and original equipment manufacturers (OEM) in the automotive industry. There are some conference papers and articles (none peer-reviewed) on the topic of hybrid processes in this context, but no scientific peer-reviewed publications [45,46,47,48,49,50]. In recent review articles, hybrid processing methods such as back-injection molding of natural fiber or wood fiber composites are not mentioned [6,7,8,9,10,11]. One industry-led project dealing with the topic of hybrid composites was the FENAFA project (completed in 2014). The project analyzed the bond strength of hybrid composites with natural fibers. A needled natural fiber non-woven with a polypropylene matrix as the semi-finished material and a natural fiber-polypropylene compound as the injection material were tested. The results showed that the bond strength decreases slightly with increasing width of the rib geometry, and that specimen failure occurs only within the natural fiber non-woven. However, the results are available only in the project report without further details on the materials or the test methods [49]. A rare scientific publication on back-injection of natural fibers is Ouali et al., who analyze the back-injection of prepregs with unidirectional flax fibers. They focus on the continuous production of prepregs with biopolymers and their mechanical behavior and only show images of back-injected structures, but do not mention their bond strength [51,52,53].

Instrumented puncture impact tests or drop weight tests are known methods to test composites and natural fiber-reinforced composites [54,55,56,57,58,59,60]. There are no articles that examine hybrid back-injected composite structures with natural or wood fibers. Pingulkar et al. review the drop weight impact characteristics of bio-composites with natural bast fibers and a thermoplastic matrix. Composites with kenaf, jute, and hemp have the potential to find application in semi-structural components [60].

## 2. Materials and Methods

### 2.1. Materials

The following table shows the two materials that were used for injection molding in test series 1 (test specimens for pull-off tests), which was carried out at the TH Rosenheim. The pull-off test is explained in more detail in Section 2.2.1. For the direct-compound in test series 1, a polypropylene (PP) homopolymer HJ120UB from Borealis was used. A polypropylene copolymer SCONA TPPP 8112 GA grafted with maleic anhydride from BYK Additives was used as a coupling agent. The direct-compound was compared with a WPC (Wood-Plastic-Composite) from the company JELU-Werk as a benchmark. The material WPC PP H50-500-14 was mixed with 60 wt.-% HJ120UB from Borealis to achieve a wood content of 20 wt.-%. Test series 2 was carried out in the technical center of FRIMO Sontra on a standard injection molding machine. Consequently, only the WPC was used in test series 2.

The spruce fibers for the direct-compound (see Table 1) and for the non-wovens (see Table 2) were produced in the technical center of the TH Rosenheim (see see Section 2.2). A needled natural fiber non-woven Hacoloft N from the company J.H. Ziegler was used as a benchmark. The benchmark was compared with a wood fiber non-woven based on air-lay technology. The wood fiber non-wovens were produced in the technical center of AUTEFA Solutions in Linz. A PP fiber of type E 4219 from IFG Asota with a length of 18 mm and a titer of 2.2 dtex was used for the wood fiber non-wovens.

### 2.2. Methods

#### 2.2.1. Test Series 1 (Test Specimen for Pull-Off Test)

Spruce logs from the Rosenheim area were manually debarked and reduced to chips using a Bruks drum chipper (type CV 400N-2M, Siwertell AB, Bjuv, Sweden). The defibering of the fresh wood chips was carried out on a 12″ laboratory refiner (type 12 1CP, Andritz AG, Graz, Austria). Thermo-mechanical pulping took place at 160 °C and 5.2 bar saturated steam pressure for 3 min. The 12XASR01 grinding disc was used for defibering. For direct compounding, the moist spruce fibers were further processed into pellets by a pelleting press (type RMP 250, MÜNCH-Edelstahl GmbH, Hilden, Germany). During pelleting, the feed rate of the metering screw and the compression are controlled manually. The wood fiber pellets were compounded with polypropylene and the coupling agent at a temperature of 170 °C on a Krauss Maffei injection-molding-compounder (type KM 300 CX IMC, KraussMaffei Group GmbH, München, Germany). Before processing the WPC, the wood fiber pellets and the coupling agent were dried at 80 °C for 3 h in a circulating air oven. After drying, the moisture content of the materials was below 0.5 wt.-%. The processing of polypropylene with wood fibers by direct compounding has been published elsewhere [12]. For the non-woven production, the wood fibers were laid with PP fibers to form a non-woven via an air-lay system. Details of this process have already been presented [61]. The non-wovens were pre-consolidated via a heated (200 °C, isobaric, type LA 100, Robert Bürkle GmbH, Freudenstadt, Germany) and a cooled press (40 °C, isochoric, type LP 370, Dieffenbacher GmbH, Eppingen, Germany) to a thickness of 2 mm (see Figure 2). Before being placed in the injection mold, the inserts were heated to a core temperature of 170 °C using an infrared oven. The infrared oven was built by the TH Rosenheim with infrared heaters from KRELUS (heater type G14 -25 -2,5 MINI 7,5, Krelus AG, Sarnen, Switzerland).

Two different rib geometries were molded (see Figure 3a,b). The first rib geometry has a radius of 0.8 mm at the rib base (see Figure 3a), which means that the interface between the rib and non-woven is 229.5 mm². The second rib geometry has a foot of 10 mm width and 1.3 mm height at the rib base, so the interface is 450 mm² (see Figure 3b). The length of both ribs is 45 mm. Each sample has eight ribs with a foot, and eight ribs with a radius.

The bond strength of the molded ribs on the non-wovens (see Figure 4a) was determined via a pull-off device. Figure 4b shows a schematic picture of the pull-off device. A Zwick/Roell type Z020 universal testing machine was used (ZwickRoell GmbH & Co. KG, Ulm, Germany). The pull-off speed was 10 mm/min.

#### 2.2.2. Test Series 2 (Test Specimen for Instrumented Puncture Impact Behavior)

Compared to test series 1, in test series 2 the unconsolidated non-wovens were heated to a core temperature of 170 °C and pressed in an isochoric process to a thickness of 2 mm via a heated press and inserted into the injection mold directly. The WPC was processed on a standard injection molding machine with a three-zone screw at 170 °C. For material testing, the central part was removed from the specimens (see Figure 5b) and tested using a puncture test according to ISO 6603-2/40/20/C/4.4 (see Figure 5c). An Amsler HIT1100F drop impact tester from ZwickRoell GmbH & Co. KG (Ulm, Germany) was used for this purpose.

#### 2.2.3. General Material Testing

Air jet sieve analyses of wood fibers were performed using an air jet sieve (type e200 LS, Hosokawa Alpine AG, Augsburg, Germany). The sieve mesh sizes used were 125, 315, 630, 1000, 1600, and 2500 μm. The sample weight was 5 g. The air jet sieve was used in combination with a scale (type PB602-S, Mettler-Toledo International Inc., Columbus, OH, USA). The melt flow rate was measured with a melt index tester (type MI-3, GÖTTFERT Werkstoff-Prüfmaschinen GmbH, Buchen, Germany) according to ISO 1133-2. The melt viscosity measurements were carried out on a high-pressure capillary viscometer (type RHEOGRAPH 25, GÖTTFERT Werkstoff-Prüfmaschinen GmbH, Buchen, Germany) according to ISO 11443. For both materials, molded samples were cut up, dried for 3 h at 80 °C in a circulating air oven, and measured afterward. Tensile testing was performed on a Zwick/Roell type Z020 tensile testing machine with a load cell of 20 kN and tactile extensometer according to ISO 527. The pull-off tests were performed on the same machine. An Amsler HIT1100F drop impact tester from Zwick/Roell was used for the puncture test by ISO 6603-2. The drop impact test was performed at 4.4 m/s with a weight of 9.378 kg at a drop height of 1 m. The microscopic images were taken with a Zeiss Smartzoom 5 digital microscope (Carl Zeiss AG, Oberkochen, Germany). The illumination allows ring light and a coaxial bright field. A microtome (type Mikrotom L, microTec Laborgeräte GmbH, Walldorf, Germany) with a linear cutting method was used for microscopy with microtome sections. Computed tomography (CT) was performed using a TomoScope XS Plus from Werth Messtechnik GmbH (Gießen, Germany).

## 3. Results and Discussion

### 3.1. Fiber Geometry

The gravimetric size distribution was measured using an air jet sieve (see Figure 6). Optical measurement methods cannot be used to compare fibers and fiber pellets, as their geometries differ too greatly. However, the low standard deviation in the analysis of spruce fibers indicates a high reproducibility. The standard deviation for the pellets is significantly higher. This is because smaller pieces detach from the pellets during screening. The spruce fibers in Figure 6 were used on the one hand for the wood fiber non-wovens, and on the other hand as pellets for direct compounding. The spruce and fir used in the WPC benchmark is very small in comparison and can be described as wood flour. The length of the natural fibers kenaf and hemp from the benchmark material is in the range of 5 to 10 cm. Consequently, no air jet sieve analysis is possible for the natural fibers.

Imken and Plinke et al. showed that the comparability between different optical methods is not always guaranteed. Dispersibility and algorithms have a great influence on the results. For the same material, measurement deviations between the methods are therefore unavoidable. Currently, there is no standard method for sufficient characterization [62,63]. A more precise analysis of the fiber geometries, as well as the length–diameter ratio (L/D ratio) via optical fiber length measuring systems (FASEP Eco System, FiberShape Cross M) and computed tomography (TomoScope XS Plus), is currently being investigated at the TH Rosenheim.

### 3.2. Viscosity

The following diagram shows the melt viscosity measurement of the two injection materials used. The MFR value of the direct-compound is (12 ± 1) g/10 min and of the WPC is (20 ± 1) g/10 min (see Table 1). The viscosity measurement with a high-pressure capillary rheometer shows that the viscosities are almost identical over a higher shear rate range (see Figure 7). The effect of the injection material on the bond strength is shown and discussed in Section 3.4. Test series 2 was carried out in the technical center of FRIMO Sontra GmbH on a standard injection molding machine. Consequently, only the WPC was used in test series 2 (see Section 3.6).

### 3.3. Effect of Non-Woven Core Temperature on the Bond Strength (Pull-Off Test)

The non-wovens were placed in the injection mold either at room temperature (20 °C) or at a core temperature of 170 °C (see Section 2.2.1 Test series 1 (test specimen for pull-off test)). Figure 8 shows the effect of two heating temperatures on both semi-finished products. It shows that by increasing the core temperature of the non-wovens from 20 °C to 170 °C before back-injection, the maximum pull-off force can be significantly increased. This applies to both semi-finished products with natural fibers and those with wood fibers. The maximum pull-off force for wood fiber non-wovens at 170 °C lags slightly behind the benchmark from the natural fiber but is in the same range. In the unheated state, there is no significant difference. The heating of the non-wovens was performed by using an infrared oven (see Section 2.2).

In the case of unheated semi-finished products, adhesive fracture failure occurs at the rib-to-non-woven interface. On the microscopic images in Figure 9, almost no fibers can be seen on the ribs after the pull-off at unheated non-wovens. In heated non-wovens, cohesive failure occurs within the non-woven, which significantly increases the maximum pull-off force. After the pull-off of the heated non-wovens in both cases, many fibers can be seen on the ribs (see Figure 9). In the case of semi-finished products with natural fibers, heating also results in a change of displacement (see Figure 8 and Figure 9). The fracture behavior in the heated state is determined by the semi-finished products and not by the interface. The much shorter wood fibers do not seem to affect this.

### 3.4. Effect of the Injection Material on the Bond Strength (Pull-Off Test)

The following diagram shows the effect of two injection materials on both semi-finished products. The injection materials have no significant effect on the maximum pull-off force within the same non-woven. This was to be expected, since the melt viscosity of both materials (see Figure 7) is very similar over the entire shear rate range. Nevertheless, the comparison is relevant, because in Section 3.4 only the WPC can be used for technical reasons. In each case in Figure 10, cohesive failure is evident within the non-wovens. In both cases, the mechanical values of wood fiber non-wovens are slightly behind the natural fiber benchmark.

### 3.5. Effect of the Rib Geometry on the Bond Strength (Pull-Off Test)

To compare both rib geometries (see Section 2.2.1) the maximum stress, instead of the maximum force, is shown in Figure 11. The maximum stress is calculated from the maximum pull-off force divided by the respective reference surface. The reference surfaces are shown in Figure 11. As a further comparison, a rectangular aluminum rib was glued to the non-wovens with cyanoacrylate. It can be seen that the ribs with radius transmit significantly higher stresses than the ribs with foot. Both rib geometries show higher values than the glued specimens. In each case, failure occurred within the non-woven. Consequently, rib-to-non-woven bond strength improves with the back-injection process. The bond strength of the ribs is higher than the pure transverse tensile strength of the non-woven (glued specimens). The improvement of the bond strength by injection depends on the rib geometry.

In any case, a cohesive fracture pattern within the non-wovens occurs. This means that the non-wovens are the weakest part of the composite. The tensile strength of the direct-compound as injection material of (42 ± 3) MPa (see Table 1) exceeds the stresses transferred to the non-woven (see Figure 11). Figure 12 shows exemplary ribs after the pull-off test.

Figure 13 shows the change in the transition area from rib to non-woven. Microtome sections of both rib geometries are shown (a,b) and the respective computed tomography images (c,d). It can be seen that, with the narrower ribs with radius (a,c), the injection pressure can act differently on the non-woven. Directly below the rib, a higher density due to the compaction is found (c), which is highlighted in blue. In both computed tomography images, the color black shows the lowest density (the background), grey and white show slightly increasing density, and blue the highest density. Below the rib with foot, only a slight increase in the density of the non-woven can be seen in the computed tomography image (d). The increase in density below the ribs leads to better bonding. Thus, higher stresses can be transmitted. Consequently, both molded-on ribs transmit higher stresses than the glued-on aluminum rib. Furthermore, the rib with radius transmits higher stresses than the rib with foot because the injection pressure can act on a smaller surface (see Figure 11). In the project FENAFA, a similar trend is found with different materials. Wider ribs led to a decrease in bond strength [49].

### 3.6. Effect of Hybridization of Non-Wovens on the Impact Behavior

The hybrid back-injection molding process can be used for functionalizing flat, semi-finished products such as non-wovens. For example, stiffening ribs and screw domes can be injected and edge areas can be molded. The following diagram shows that back-injection with WPC can also improve the impact behavior.

In test series 2, unconsolidated non-wovens were heated to a core temperature of 170 °C via a calibration press and inserted into the injection mold directly. For material testing, the central part of the specimen was removed, and tested using a puncture test (see Figure 5b). Figure 14 shows the maximum impact force and the penetration energy of the pure non-woven and the hybrid of the respective inserts. It can be seen that the non-wovens with ribs (hybrid) can absorb significantly higher maximum impact forces and energies compared to the non-wovens without ribs.

The improving maximum force and energy during multiaxial impact testing are good indicators of bond strength under impact stress. Figure 15 shows an exemplary curve of each sample. The curve progression of force and displacement of the pure non-woven is superimposed by the ribs in the hybrid. The ribs lead to a higher maximum impact force and higher energy absorption with similar deformation. The benchmark with natural fibers shows slightly higher values (see Figure 15a) compared to the specimens with wood fibers (see Figure 15b).

The following figure shows the respective test specimens after the impact test. In the case of the non-wovens a clean puncture can be seen (see Figure 16a,b). In the case of the hybrids, the fracture behavior is very brittle (see Figure 16c,d). Nevertheless, the maximum forces absorbed and the energy absorbed is significantly higher with the hybrids (see Figure 14 and Figure 15).

## 4. Conclusions

This article compares the bond strength of non-wovens with wood fibers (spruce) and alternatively non-wovens with natural fibers (kenaf, hemp) in thermoplastic hybrid composites. The comparison uses wood fiber non-wovens and natural fiber non-wovens with a polypropylene matrix. The non-wovens are back-injected with direct-compounded wood-fiber-reinforced polypropylene using an injection-molding-compounder.

Thermoplastic hybrid composites with wood fiber non-wovens can compete with the benchmark made of natural fibers. Hybrid composites with wood fibers benefit from the synergy of the combination of both compression and injection molding. Back-injection molding combines the inherent stiffness of fiber-reinforced semi-finished products in compression molding with the design stiffness and high productivity of injection molding [44]. Hybrid composites with wood fibers show slightly lower, but quite comparable, values. It is shown that the core temperature and rib geometry greatly affect the bond strength of back-injected ribs. At a core temperature of 170 °C, the pull-off force can be increased from around 800 N to almost 1400 N for hybrid composites with wood fiber non-wovens. Both injection materials showed no significant difference regarding the bond strength. Back-injection with WPC can also improve the impact behavior of the composite. With a direct-compound or WPC, it is possible to manufacture hybrid composites as a single-material solution with domestic raw materials, because for non-wovens and the injection material the same wood fiber can be used. Thermoplastic hybrid composites with wood fiber non-wovens present, therefore, an interesting alternative to natural fibers in technical semi-structural applications. Potential growing markets for biocomposites are primarily the construction sector, the automotive sector as well as small new electric car manufacturers [5].

## Figures and Tables

**Figure 1 materials-15-02473-f001:**
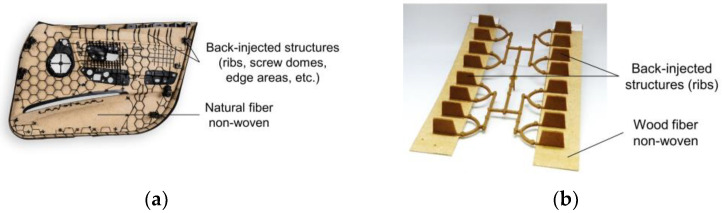
(**a**) State-of-the-art technology door panel from automotive interior made of a natural fiber non-woven with back-injected structures (ribs, screw domes, edge areas, etc.), image from Yanfeng Automotive Interiors with added labeling [16]; (**b**) test specimen for analyzing the bond strength via pull-off test, for a non-woven made of wood fibers (spruce) and PP, with the ribs back-injected onto the non-woven.

**Figure 2 materials-15-02473-f002:**
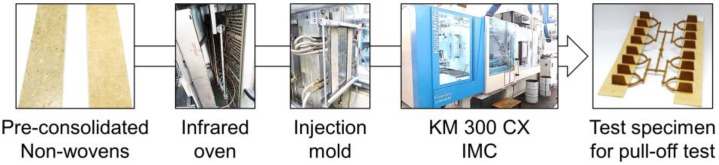
Process flow from pre-consolidated non-wovens to test specimen for pull-off tests.

**Figure 3 materials-15-02473-f003:**
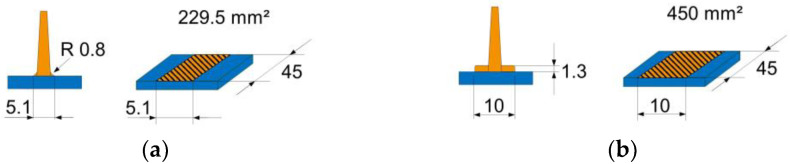
Two different rib geometries were molded for investigating the bond strength. (**a**) rib with radius; (**b**) rib with foot.

**Figure 4 materials-15-02473-f004:**
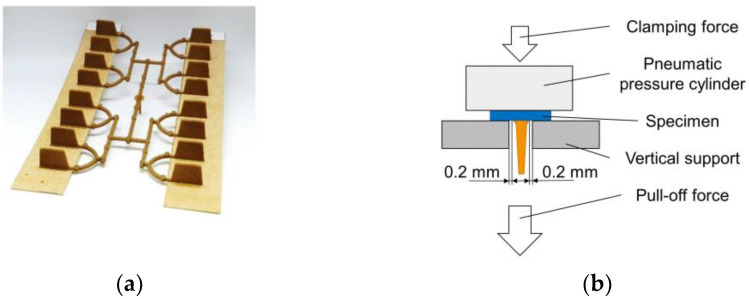
Test setup for investigating the bond strength. (**a**) Test specimen for pull-off test, wood fiber non-woven with both molded rib geometries; (**b**) schematic picture of the pull-off device.

**Figure 5 materials-15-02473-f005:**
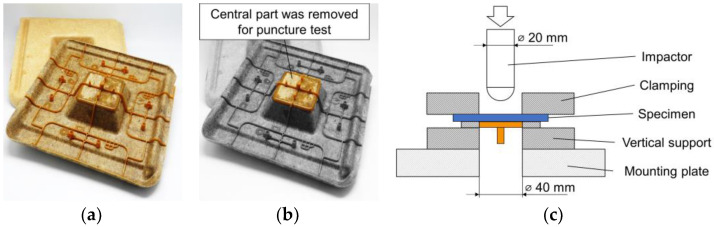
Test setup for investigating the puncture impact behavior according to ISO 6603-2/40/20/C/4.4. (**a**) Test specimen with natural fiber non-woven (front) and specimen with wood fiber non-woven (back); (**b**) central part (highlighted in color) was removed for puncture test; (**c**) schematic picture of the puncture impact behavior test.

**Figure 6 materials-15-02473-f006:**
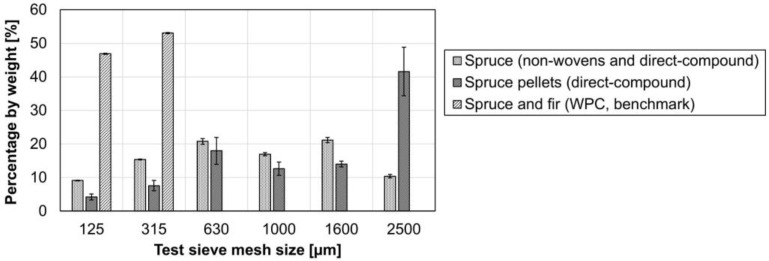
Air jet sieve analysis of wood fibers from spruce, their pellets, and the wood flour used in the benchmark WPC; *n* = 3 ± SD (standard deviation).

**Figure 7 materials-15-02473-f007:**
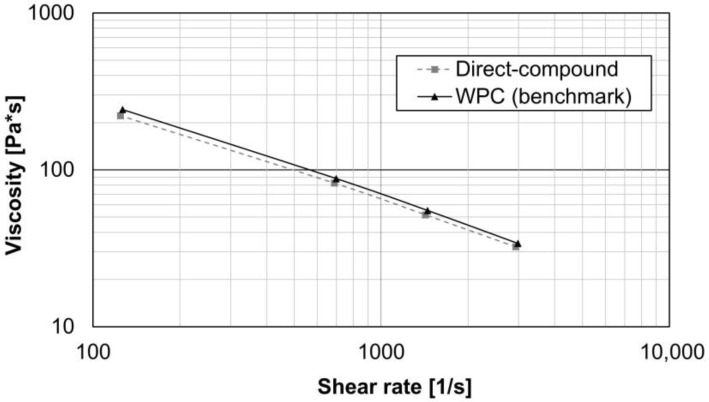
Melt viscosity of both injection materials according to ISO 11443 at 200 °C. For both materials, molded samples were cut up, dried for 3 h at 80 °C in a circulating air oven, and then measured.

**Figure 8 materials-15-02473-f008:**
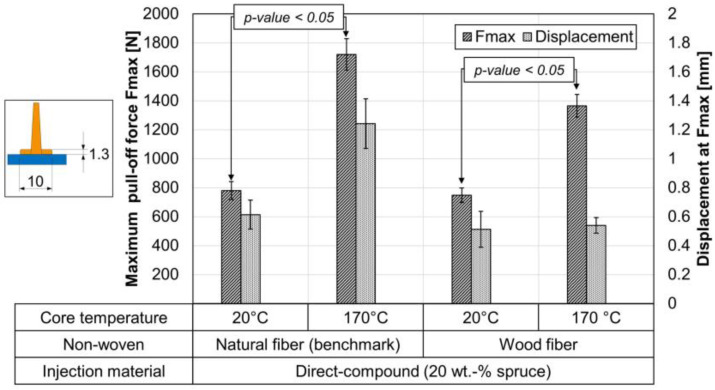
Pull-off test of ribs with foot at two different non-woven core temperatures; injection material direct-compound with 20 wt.-% spruce; *n* = 7 ± SD; *p*-value with single-factor ANOVA at α = 0.05.

**Figure 9 materials-15-02473-f009:**
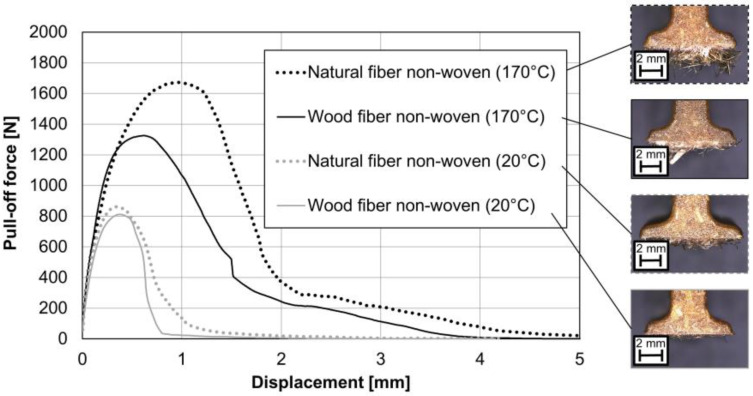
Pull-off test of ribs with foot at two different non-woven core temperatures; exemplary curve progression of the force-displacement diagram and microscopic images after pull-off.

**Figure 10 materials-15-02473-f010:**
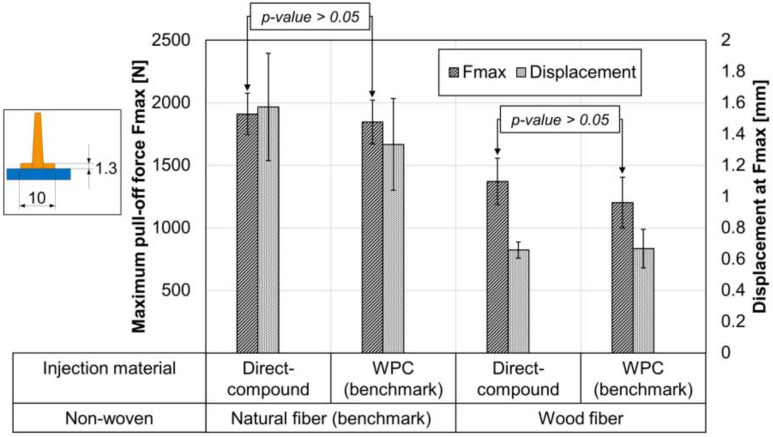
Pull-off test of ribs with foot and two injection materials; non-woven core temperature at 170 °C before inserting into the mold; *n* = 7 ± SD; *p*-value with single-factor ANOVA at α = 0.05.

**Figure 11 materials-15-02473-f011:**
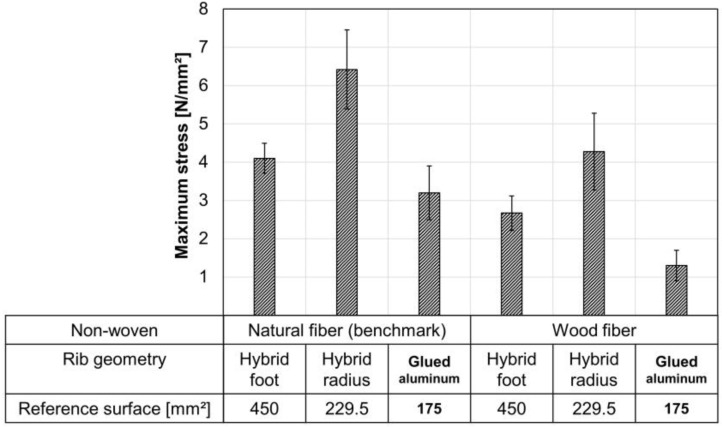
Pull-off test with two rib geometries and both non-wovens; non-woven core temperature at 170 °C before inserting into the mold; *n* = 5 ± SD; injection material direct-compound 20 wt.-% spruce.

**Figure 12 materials-15-02473-f012:**
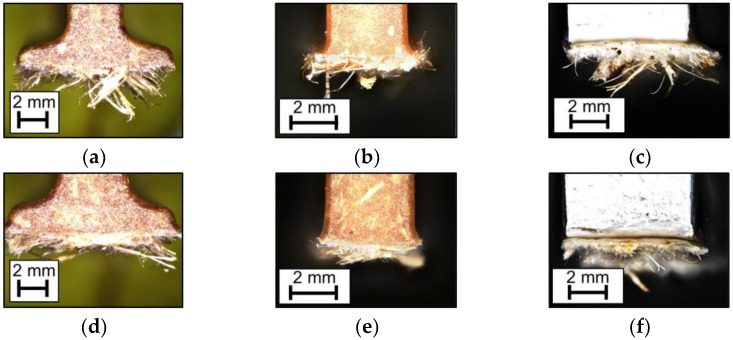
Exemplary ribs after pull-off from Figure 11, reflected light microscopy; (**a**–**c**) ribs with natural fiber non-wovens; (**d**–**f**) ribs with wood fiber non-wovens. (**a**,**d**) ribs with foot; (**b**,**e**) ribs with radius; (**c**,**f**) glued-on aluminum ribs.

**Figure 13 materials-15-02473-f013:**
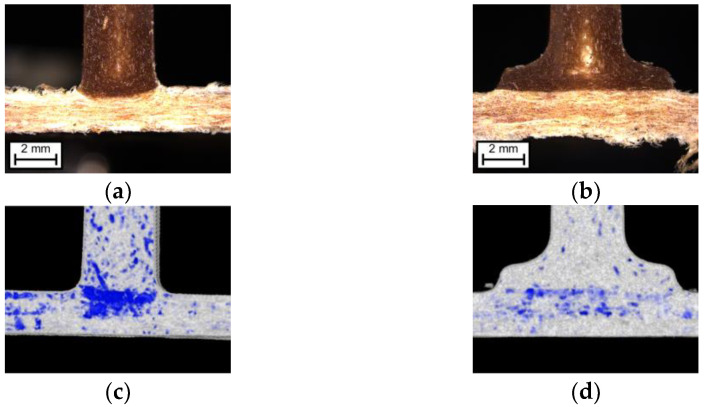
Wood fiber non-woven injected with direct-compound: (**a**) rib with radius microtome section, reflected light microscopy; (**b**) rib with foot microtome section, reflected light microscopy; (**c**) rib with radius, computed tomography; (**d**) rib with foot, computed tomography. In both computed tomography images, black shows the lowest density (background), grey and white slightly increasing density, and blue the density.

**Figure 14 materials-15-02473-f014:**
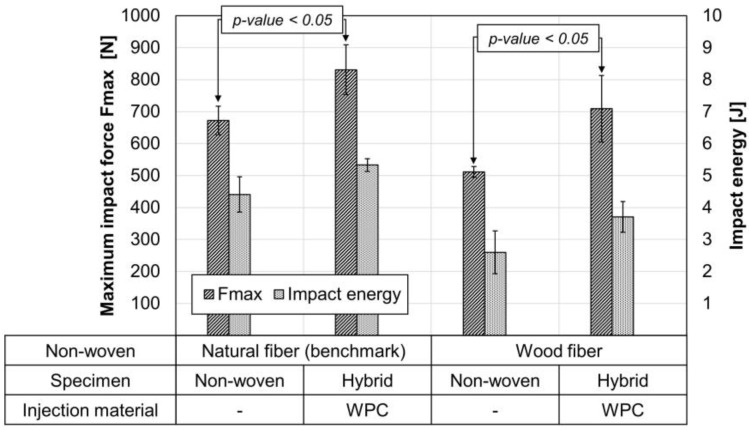
Instrumented puncture impact behavior according to ISO 6603-2/40/20/C/4.4; v = 4.4 m/s; m = 9.378 kg; h = 1 m; with lubrication; *n* = 3 ± SD. Both materials were tested as non-wovens and as hybrid with ribs; *p*-value with single-factor ANOVA at α = 0.05.

**Figure 15 materials-15-02473-f015:**
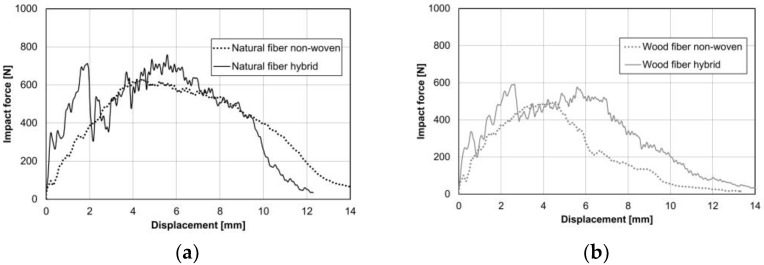
Instrumented puncture impact behavior according to ISO 6603-2/40/20/C/4.4; v = 4.4 m/s; m = 9.378 kg; h = 1 m; with lubrication; exemplary curves. Both materials were tested as non-wovens and as hybrid with rib; (**a**) natural fiber non-woven and hybrid; (**b**) wood fiber non-woven and hybrid.

**Figure 16 materials-15-02473-f016:**
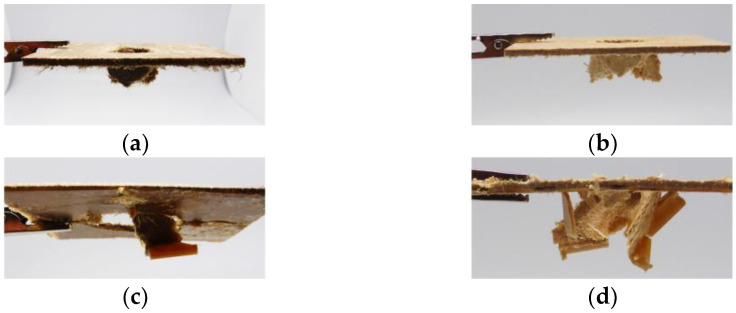
Exemplary specimen after impact test. (**a**) natural fiber non-woven; (**b**) wood fiber non-woven; (**c**) natural fiber non-woven with reinforcement rib (hybrid); (**d**) wood fiber non-woven with reinforcement rib (hybrid).

**Table 1 materials-15-02473-t001:** Injection material for test series 1 and 2. In test series 2, only the WPC was used. The tensile modulus and tensile strength were measured according to ISO 527-2/1A, the MFR (melt flow rate) was measured according to ISO 1133-2 with 10 kg at 170 °C.

	WPC (Benchmark)	Direct-Compound
Wood fibers	Spruce mixed with fir (20 wt.-%)	Spruce (20 wt.-%)
Polymer	PP homopolymer (78.8 wt.-%)	PP homopolymer (77 wt.-%)
Additive	Coupling agent (1.2 wt.-%)	Coupling agent (3 wt.-%)
Tensile modulus	(3010 ± 50) MPa	(2940 ± 210) MPa
Tensile strength	(39 ± 1) MPa	(42 ± 3) MPa
MFR	(20 ± 1) g/10 min	(12 ± 1) g/10 min

**Table 2 materials-15-02473-t002:** Non-wovens for test series 1 and 2. The tensile modulus and strength were measured according to ISO 527-4/2.

	Natural Fiber Non-Woven (Benchmark)	Wood Fiber Non-Woven
Fiber	Kenaf, hemp (50 wt.-%)	Spruce (50 wt.-%)
Polymer	PP-fiber	PP-fiber
Tensile modulus [MPa]	2900 ± 200	2950 ± 60
Tensile strength [MPa]	28 ± 2	28 ± 2
Surface weight [g/m²]	1800 ± 50	1920 ± 40

## Data Availability

The data presented in this study are available on request from the corresponding author.

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
