# Peer review of "Thermoplastic Hybrid Composites with Wood Fibers: Bond Strength of Back-Injected Structures"

_materials, 2022, doi:10.3390/ma15072473_

Round 1
Reviewer 1 Report
1. In this paper, only polypropylene, a kind of plastic, is used, so the title provided by the author is too large, so it is suggested to modify it. 2. It is necessary to supplement the experimental data in order to improve the persuasion. 3. There are as many as 10 keywords. It is suggested to delete keywords with less relevance. 4. The preface is too trembling, so the author is suggested to simplify it. 5. The subtitle of the results and discussion section is suggested to be modified appropriately. 6. It is suggested to supplement the micro morphology of the pull off part. 7. Please add how to embody the advantages of back injected structures.Author Response
Please see the attachment

Reviewer 2 Report
General Comments to the Authors
General aim and scope of this manuscript entitled “Thermoplastic hybrid composites with wood fibers: bond strength of back-injected structures” seems to be appropriate for “Materials”. The authors produced natural fiber reinforced thermoplastic hybrid composites by back-injection molding. The authors should improve quality of the manuscript before publication.
Specific comments are as follows:
More literature search should be done on natural materials / polymer biocomposites and summarize in the Introduction section of this manuscript.
It is highly recommended that the authors should read and summarize the below articles into the manuscript.
- Mohanty, A. K., Misra, M., and Hinrichsen, G. 2000. Biofibres, biodegradable polymers and biocomposites: An overview, Macromol. Mater. Eng. 276-277(1), 1-24.
- Ayrilmis, N., Akbulut, T., Dundar, T., White, R.H., Mengeloglu, F., Buyuksari, U., Candan, Z., Avci, E. 2012. Effect of boron and phosphate compounds on physical, mechanical, and fire properties of wood-polypropylene composites. Construction and Building Materials 33: 63 – 69.
- Candan, Z., Gardner, D. J., Shaler, S. M. 2016. Dynamic mechanical thermal analysis (DMTA) of cellulose nanofibril/nanoclay/pMDI nanocomposites. Composites Part B: Engineering 90: 126 – 132.
Photographs of the raw materials, experimental set up (production), and final composites should be added into the manuscript.
More data from the analysis should be indicated in the Abstract.
What is the moisture content values of the wood fiber?
What about the homogeneity of the mix?
Statistical analysis should be supplied.
Some conclusions and suggestions regarding with industrial perspective should be added into both Abstract and Conclusions section of the manuscript.
Reviewer 3 Report
In the present manuscript the authors describe their study on thermoplastic hybrid composites with wood fibers, mainly concerning the bond strength of back-injected stuctures. The work is well presented and the results and conclusions are supported by the experimental data. The fact that thermoplactic hybrid polymers can be used as an alternative to natural fibers is of interest for the scientific community of composite materials and their applications. I think that the manuscript can be accepted for publication.
Reviewer 4 Report
This article compares the bond strength of non-wovens with wood fibers (spruce) 340 and alternatively non-wovens with natural fibers (kenaf, hemp) in thermoplastic hybrid 341 composites. The content of paper is good. The results are interesting and the experimental work is well conducted. I have some comments before accepting this paper for publication:
1- The English language should be revised.
2-The Abstract part should be summarized.
3- The conclusion part should contain the core findings of the work ( data)
4- The purity of all used chemicals should be mentioned in the experimental part.
5- The authors should cite the following papers in the introduction part to show the importance of hybrid composites in different fields: Ionics 25 (2019) 2645–2656 ;Journal of Molecular Liquids 260C (2018) 237-244
6-The authors should include a comparison with similar composites to show the novelty of this work.
Round 2
Reviewer 1 Report
The author carefully revised the manuscript and agreed to accept it for publication.
Reviewer 4 Report
The authors have addressed all comments in a good manner. I recommend publication of this paper in materials journal.